# Subacute Pulmonary Toxicity of Glutaraldehyde Aerosols in a Human In Vitro Airway Tissue Model

**DOI:** 10.3390/ijms232012118

**Published:** 2022-10-11

**Authors:** Yiying Wang, Qiangen Wu, Baiping Ren, Levan Muskhelishvili, Kelly Davis, Rebecca Wynne, Diego Rua, Xuefei Cao

**Affiliations:** 1Division of Genetic and Molecular Toxicology, National Center for Toxicological Research, U.S. Food and Drug Administration, Jefferson, AR 72079, USA; 2Division of Biochemical Toxicology, National Center for Toxicological Research, Jefferson, AR 72079, USA; 3Toxicologic Pathology Associates, Jefferson, AR 72079, USA; 4Division of Biology, Chemistry and Materials Science, Office of Science and Engineering Laboratories, Center for Devices and Radiological Health, Silver Spring, MD 20993, USA

**Keywords:** air-liquid-interface (ALI) airway model, glutaraldehyde (GA), airway toxicity, cloud liquid aerosol generation and exposure system, medical device development tools

## Abstract

Glutaraldehyde (GA) has been cleared by the Center for Devices and Radiological Health (CDRH) of the Food and Drug Administration (FDA) as a high-level disinfectant for disinfecting heat-sensitive medical equipment in hospitals and healthcare facilities. Inhalation exposure to GA is known to cause respiratory irritation and sensitization in animals and humans. To reproduce some of the known in vivo effects elicited by GA, we used a liquid aerosol exposure system and evaluated the tissue responses in a human in vitro airway epithelial tissue model. The cultures were treated at the air interface with various concentrations of GA aerosols on five consecutive days and changes in tissue function and structure were evaluated at select timepoints during the treatment phase and after a 7-day recovery period. Exposure to GA aerosols caused oxidative stress, inhibition of ciliary beating frequency, aberrant mucin production, and disturbance of cytokine and matrix metalloproteinase secretion, as well as morphological transformation. Some effects, such as those on goblet cells and ciliated cells, persisted following the 7-day recovery period. Of note, the functional and structural disturbances observed in GA-treated cultures resemble those found in ortho-phthaldehyde (OPA)-treated cultures. Furthermore, our in vitro findings on GA toxicity partially and qualitatively mimicked those reported in the animal and human survey studies. Taken together, observations from this study demonstrate that the human air-liquid-interface (ALI) airway tissue model, integrated with an in vitro exposure system that simulates human inhalation exposure, could be used for in vitro-based human hazard identification and the risk characterization of aerosolized chemicals.

## 1. Introduction

Ortho-phthaldehyde (OPA), peracetic acid, and glutaraldehyde (GA) are among the high-level disinfectants currently cleared by the Center for Devices and Radiological Health (CDRH) of the US Food and Drug Administration (FDA) and have been used for disinfecting heat-sensitive medical equipment (e.g., endoscopes and microsurgical instruments) in hospitals and healthcare facilities. A complete list of the FDA-cleared liquid chemical sterilants and high-level disinfectants is available for consultation [1].

GA, or 1,5-pentanedial, is an aliphatic dialdehyde with a well-studied in vivo respiratory toxicity profile [2]. Its high reactivity towards proteins makes it a widely used cross-linking agent. GA is also used as a disinfectant owing to its alkylating mode of microbicidal action, but such use has been curtailed due to its known dose-dependent skin and respiratory sensitizing effects [3]. Moreover, in both humans and animals, dose-dependent irritation of the upper respiratory tract from exposure to GA aerosols has been reported [4]. It is likely, but currently undetermined, whether the features of GA that confer its useful microbicidal properties may also be responsible for its irritating and allergenic properties.

Despite the fact that the healthcare industry has used GA for many decades to disinfect and sterilize medical devices, its effects on the human respiratory system have not been thoroughly studied due to the lack of methodology to perform the relevant testing. For example, there are only limited data on the modulation of cytokine secretion in the respiratory system as a result of GA exposure. The effects of GA exposure on other responses, such as mucociliary clearance (MCC), an essential defense mechanism in the airways [5], are also unknown. Such knowledge is important for comprehensively evaluating the health risks in humans posed by GA, given that dysregulation of cytokine production and MCC can contribute to the pathological changes in a multitude of respiratory diseases, e.g., asthma, chronic obstructive pulmonary disease (COPD), and various cough syndromes [6,7]. With the advancement of the in vitro human air-liquid-interface (ALI) airway epithelial tissue models, these key data gaps, at least in a qualitative way, can now be filled [8].

Fully differentiated ALI airway cultures consist of beating ciliated cells, mucus-producing goblet cells, and progenitor basal cells, the three major cell types found in lower large airways, arranged in a pseudostratified structure [9,10]. The functional and structural resemblance of this model to the epithelial lining of the in vivo human airway makes possible the measurement of respiratory disease-relevant toxicology endpoints, such as changes in mucin secretion, cilia beat frequency (CBF), oxidative status, cytokine modulation, and tissue morphology. Furthermore, these endpoints can be more conveniently measured in the ALI cultures than in animals and humans [10,11,12]. Thus, the ALI airway model represents a simplified yet physiologically relevant biological platform that can not only assess the respiratory toxicity of airborne chemicals, but also dissect the molecular mechanisms underlying their toxicity. In the current study, we employed the VITROCELL^®^ Cloud Liquid Aerosol Exposure System (Waldkirch, Germany) and exposed the ALI airway models to GA aerosols in a manner that both simulates in vivo aerosol exposures and allows dosimetry measurements of the test substance. A repetitive treatment regimen was used to explore the potential subacute adverse respiratory effects from occupational exposure to GA aerosols.

## 2. Results

### 2.1. Quantification of Glutaraldehyde (GA) Aerosol Deposition

Previously, our laboratory demonstrated the repeatability of test substance nebulization and the spatial deposition homogeneity of the Cloud Liquid Aerosol Generation System for dihydroxyacetone [10] and OPA [12]. Thus, a simplified validation protocol consisting of measurement at three representative base module positions from three independent nebulizations was employed in this study.

The deposition of GA aerosols increased proportionally with increasing stock solution concentration (Figure 1A). Average deposition concentrations of 0.5, 0.8, and 1.9 µg/cm^2^ (hereafter referred to as low, mid, and high concentrations) were obtained with 0.5, 0.75, and 1.0 mg/mL GA stock solutions, respectively. The inter-day coefficient of variation (CV) was within 10% for the low and high concentration groups, and within 20% for the mid concentration group. The intra-day CVs, i.e., the variation between positions within the base module, were generally within 10%, except for 2 nebulizations (nebulization with 0.5 mg/mL stock solution on day 1: %CV is ~13%; nebulization with 0.75 mg/mL stock solution on day 3: %CV is ~11%). Deposition fractions (DFs), an indicator for the efficiency of GA aerosolization, were approximately 53%, 61%, and 70% for the 0.5, 0.75, and 1.0 mg/mL GA stock solutions, respectively (Figure 1B). In general, reproducible, uniform depositions of GA aerosols for all three test concentrations were achieved with the VITROCELL^®^ Cloud System.

### 2.2. Cytotoxicity of GA Aerosols in ALI Cultures

The cytotoxicity of GA aerosols was assessed by measuring the activity of LDH released into the apical wash (Figure 2A, upper panel) and basolateral medium (Figure 2A, lower panel) 24 h after one (T1) and five exposures (T5), as well as 7 days after the last exposure (PT7). No changes in LDH activity were observed in the low and mid concentration groups at all three timepoints. The repeated exposure of ALI cultures to the high concentration of GA aerosols, however, elicited marked increases in the activity of released LDH in both the apical and basolateral compartments. The levels of LDH activity in this group returned to the baseline at PT7. Contrary to the significant increase in LDH activity by the high concentration of GA aerosols, cell viability measured by the MTS tetrazolium assay was minimally affected by all three test concentrations at T5 (data not shown).

Repeated exposure to GA aerosols induced apoptosis in a concentration-dependent manner at T5, as reflected by the increases in the percentage of cleaved caspase-3-positive cells, also called the apoptotic index (AI) (Figure 2B, T5). An approximate five-fold increase in AI was observed in the high concentration group at T5. AI was also higher in the low and mid concentration groups than that in the PBS-exposed vehicle control group, although it did not reach statistical significance. After a 7-day recovery, AI in the low and mid concentration groups was comparable to that in the vehicle control group (Figure 2B, PT7). Unexpectedly, AI in the high concentration group was about 50% lower than the vehicle control group; however, statistical significance was not achieved, likely due to the large variation observed in these samples. The percentage of Ki-67-positive cells (PI) exhibited a concentration-dependent increase after five repeated exposures (Figure 2B, T5). Even though these changes in PI were not statistically significant, an approximate twofold increase at the high concentration group may be biologically significant. PI in all treatment groups returned to the baseline after a 7-day recovery (Figure 2B, PT7).

### 2.3. Reversible Disruption of Glutathione Homeostasis by GA Aerosols

Exposing ALI cultures to the high concentration of GA aerosols led to an approximate 60% decrease in intracellular GSH levels shortly after the first treatment (Figure 3A, 20 min); the levels of GSSG remained unaffected at this timepoint (Figure 3B, 20 min). As a result, a decreasing trend in GSH/GSSG ratios was observed (Figure 3C, 20 min). The intracellular levels of both GSH and GSSG increased in a concentration-dependent manner at T1 and T5 (Figure 3A,B). As the relative changes in GSH and GSSG levels were comparable, the GSH/GSSG ratios remained relatively stable (Figure 3C, T1 and T5). After a 7-day recovery, the glutathione homeostasis generally returned to baseline levels (Figure 3A–C, PT7).

### 2.4. Modulation of Select Oxidative Stress Markers by GA Aerosols

The effect of GA aerosol exposure on the protein expression of heme oxygenase 1 (HMOX-1), NAD(P)H quinone dehydrogenase 1 (NQO1), and glutamylcysteine synthetase (GCS) was measured by immunoblotting at T1, T5, and PT7. A single exposure to both the mid and high concentrations of GA aerosols markedly upregulated HMOX-1 protein expression (Figure 4, T1, quantification shown in the right panel). The extent of induction was comparable between T1 and T5 (Figure 4, T1 and T5). HMOX-1 expression in GA-exposed groups returned to baseline levels after a 7-day recovery (Figure 4, PT7). 

The expression of NQO-1 and GCS demonstrated a biphasic response during the exposure phase. GA aerosols inhibited the expression of both proteins by approximately 50% 24 h after the first exposure (Figure 4, T1). Repeated exposure to GA aerosols reversed the inhibitory effects on their expression (Figure 4, T5). In fact, the protein levels of NQO1 and GCS in the mid and/or high concentration groups were slightly higher (by approximately 20% to 30%) than that in the vehicle control. At the end of a 7-day recovery, the expression of both proteins in the GA-exposed groups was comparable to that in the PBS-exposed vehicle control (Figure 4, PT7). 

### 2.5. Modulation of Cytokine and Matrix Metalloproteinases (MMP) Secretion by GA Aerosols

Temporal secretion of key cytokines and MMPs was quantified in the basolateral medium at T1, T5, and PT7. Exposure to GA aerosols modulated the secretion of several cytokines and MMPs in a time-dependent manner (Table 1). Mostly marginal effects on cytokine secretion were noted in cultures dosed with the low and mid concentrations of GA aerosols. A single exposure to the high GA concentration induced the secretion of IL-6, IL-8, and IL-9 by approximately 50%, 70%, and 25%, respectively, and decreased the secretion of PDGF-ΒB by approximately 35% (Table 1, T1). The stimulatory effects of GA aerosols on IL-8 and IL-9 were sustained at T5, while the inhibition of PDGF-ΒB secretion and induction of IL-6 secretion diminished after five repeated GA exposures. The secretion of VEGF was greatly decreased at this timepoint (Table 1, T5). At the end of a 7-day recovery, the release of these cytokines was mostly decreased, regardless of their responses during the exposure phase (Table 1, PT7).

An unexpected, yet intriguing, observation is the delayed modulation of cytokine release by GA aerosols. Six cytokines (i.e., IL-1RA, FGF basic, G-CSF, GM-CSF, MCP-1, and TNF-α), whose secretion was not altered during the exposure phase, were detected at levels significantly different than that in the PBS-exposed vehicle control after a 7-day recovery (Table 1, PT7). Specifically, the secretion of IL-1RA was markedly enhanced, whereas the release of the rest of the five cytokines was decreased at PT7.

The effects of GA aerosols on MMP secretion were relatively modest. Among the eight MMPs screened in this study, only the secretion of MMP-12 was stimulated by approximately 50% after five repeated exposures to the high concentration of GA aerosols (Table 1, T5). After a 7-day recovery, instead of returning to the baseline levels, the secretion of MMP-12 in this group was decreased by approximately 70% (Table 1, PT7). Similar to the residual effects of GA aerosols on cytokine secretion, the levels of MMP-1 and MMP-7, whose release was not modulated during the exposure phase, were found to be significantly lower when compared to those in the PBS-exposed vehicle controls at PT7.

### 2.6. Effects of GA Aerosols on Ciliary Cells in ALI Cultures

GA aerosols suppressed CBF in a concentration- and time-dependent manner during the exposure phase. A single exposure to the high concentration of GA aerosols significantly decreased CBF (Figure 5A). The ciliostatic effects of GA aerosols appeared to be cumulative, as repeated exposures to the mid and high concentrations of GA aerosols completely eliminated CBF in a temporal manner (Figure 5A, T3 and T5). CBF in the mid concentration group returned to the baseline frequency after a 7-day recovery; CBF in the high concentration group, however, remained undetectable at PT7 (Figure 5A, PT7).

The expression of four ciliary proteins was measured at T1, T5, and PT7 (Figure 5B, quantification shown in the right panel). The expression of acetylated α-tubulin and DNAI1 was decreased, whereas the expression of CDC20B and FoxJ1 was increased during the exposure phase. The effect of GA aerosols on the expression of acetylated α-tubulin was relatively moderate. A slight but statistically significant reduction in its expression was observed in all three concentration groups after a single exposure (Figure 5B, T1). Its expression in the high concentration group remained lower than that in the PBS-exposed vehicle control at T5 and PT7 (Figure 5B, T5 and PT7). Compared to the effect of GA aerosols on acetylated α-tubulin expression, its inhibitory effect on DNAI1 expression was more potent. A single exposure decreased the expression of DNAI1 by approximately 50% in all concentration groups (Figure 5B, T1). Its expression in the low and mid concentration groups appeared to be alleviated after five repeated exposures, while its expression level in the high concentration group remained at a lower level at T5 (Figure 5B, T5). The expression of DNAI1 returned to baseline levels after a 7-day recovery (Figure 5B, PT7). 

Multiple exposures to GA aerosols were necessary to upregulate the expression of CDC20B (Figure 5B, T5). The stimulatory effect of GA aerosols on CDC20B expression was reversible in the low- and mid-concentration groups; its expression in the high concentration remained elevated at the end of a 7-day recovery (Figure 5B, PT7). The effect of GA aerosols on FoxJ1 expression was more immediate. A single GA aerosol exposure greatly increased the expression of FoxJ1 in the mid- and high-concentration groups; the expression of FoxJ1 remained upregulated throughout the exposure phase and decreased towards baseline levels at PT7 (Figure 5B).

### 2.7. Disruption of Mucin Homeostasis by GA Aerosols in ALI Cultures

Repeated exposure to GA aerosols inhibited the secretion and expression of MUC5AC, MUC5B, and CCSP (Figure 6A,B, T5). Decreases in MUC5AC and MUC5B secretion occurred at T5 for the mid and high GA concentrations. A 7-day recovery alleviated the inhibitory effects of GA exposure for the mid concentration but not for the high concentration group (Figure 6A, PT7). The inhibition of CCSP secretion was observed only in the high concentration group after five repeated exposures (Figure 6A, T5). Its secretion completely returned to the baseline after a 7-day recovery (Figure 6A, PT7).

Five consecutive exposures to the mid and high concentrations of GA aerosols downregulated the expression of all three mucin proteins (Figure 6B, T5). After a 7-day recovery, their expression returned to levels comparable to that in the vehicle control for the mid concentration group (Figure 6B, PT7). The inhibition seen for the high concentration group was only slightly alleviated and remained significantly lower than the baseline expression. 

Exposure to the low, mid, and high concentrations of GA aerosols for five consecutive days caused the minimal, mild, and moderate depletion of goblet cells, respectively (Table 2, T5). Cultures exposed to the high GA concentration exhibited disrupted epithelium on the apical surface. Such disruption is characterized by the stratification and shedding of epithelial cells, which could have led to a decreased number of goblet cells compared to the PBS-exposed vehicle controls at T5. This microscopic finding was corroborated by the histopathologic staining for goblet cells (Figure 6C, T5). The quantification of goblet cell density revealed significant reductions in both the mid and high concentration groups. The overall tissue morphology of the ALI cultures in these two groups were partially restored at the end of a 7-day recovery; goblet cell density was slightly increased at PT7, but was still significantly lower than that in the vehicle control group (Figure 6C, PT7).

### 2.8. Effect of GA Aerosols on General Morphology in ALI Cultures

The morphology of the ALI cultures was evaluated in H&E-stained tissue sections prepared from cultures given five repeated daily exposures to GA aerosols and those after five daily exposures plus a 7-day recovery (Table 2). In general, the tissue morphology was minimally affected by the low and mid concentrations of GA aerosols; multiple non-neoplastic lesions, i.e., necrosis, atrophy, decreased ciliation, goblet cell depletion, and squamous differentiation, were observed in the high concentration group. Minimal to mild necrosis consisting of sloughed individual flattened (squamoid) to polygonal epithelial cells was found on the apical surface of cultures exposed to the high concentration of GA aerosols at T5; the necrotic effect of GA aerosols was reversed after a 7-day recovery. Cultures exposed to the high GA concentration also exhibited minimal to mild atrophy characterized by a decrease in the thickness of the airway tissue and the presence of cuboidal, low columnar and/or flattened cells accompanied by the loss of tissue organization in certain areas. The atrophic effect, however, persisted at PT7.

Consistent with the ciliary protein immunoblotting results, decreased ciliation was observed and considered related to repeated exposures to the mid and high concentrations of GA aerosols. An increased number of epithelial cells with no cilia, fewer cilia, and/or cilia of decreased length were found on the apical surface of the treated cultures. 

Repeated exposure to the high concentration of GA aerosols induced minimal to mild squamous differentiation at T5. The affected areas were located at or near the apical surface of the cultures and consisted of polygonal to spindle-shaped (squamoid) cells with intercellular bridges and eosinophilic cytoplasm. IHC staining with involucrin, a marker for squamous differentiation, corroborated the pathology observations. Involucrin staining tended to be enriched at or near the apical surface; after five consecutive exposures, staining intensities were significantly higher in the high concentration group compared to the vehicle control group (Figure 7A, T5). The intensity of involucrin staining decreased but was still stronger than the PBS-exposed vehicle control at PT7 (Figure 7A, PT7). A similar pattern of involucrin expression was observed using immunoblotting (Figure 7B). However, based on the morphology of the H&E-stained slides, the epithelium was devoid of squamous differentiation at PT7 (Table 2). The effect of GA aerosol exposures on tissue permeability of the ALI cultures was evaluated by measuring TEER at T5 and PT7. TEER increased in the mid and high concentration groups following five daily exposures and remained high in the high concentration group at PT7 (Figure 7C).

## 3. Discussion

In this study, we showed that repeated exposure to glutaraldehyde (GA) aerosols induced a panel of temporal changes in tissue responses in the ALI cultures. In general, the toxicity responses to GA exposure in the ALI airway cultures are consistent with observations on other aldehydes, i.e., ortho-phthalaldehyde (OPA), formaldehyde, and acrolein, as reported in our previous studies [11,12,13].

Glutathione (GSH) is a major cellular antioxidant, protecting cells from oxidative damages and maintaining redox homeostasis [14]. An immediate depletion of intracellular GSH in the absence of changes in GSSG levels in the treated cultures suggested that GSH depletion may be due to mechanisms other than GSH oxidative conversion to GSSG. One possibility is the direct binding of GA with the SH-group of GSH, leading to a prompt reduction in free GSH [15,16]. Additionally, GA has been shown to decrease the activity of glutathione reductase and glucose-6-phsophate dehydrogenase in human erythrocytes [17]. In this study, the expression of GCS, the first and rate-limiting enzyme involved in the de novo GSH synthesis [18], was found to be downregulated by GA exposure. Modulation of these enzymes all occurred after a single treatment with GA. Thus, the combinatorial inhibition of these enzymes by GA may have delayed the timely replenishment of intracellular GSH in the treated cultures. Both mechanisms could be at work simultaneously. With prolonged GA exposures, both GSH and GSSG levels increased at comparable magnitudes, suggesting that the restoration of GSH biosynthesis, possibly as a result of an adaptive cellular response (e.g., GCS upregulation at T5), triggered the re-establishment of the GSH/GSSG redox balance in GA-exposed cultures.

NQO1 and GCS, which are involved in generating antioxidant potential [18,19], were significantly inhibited after one exposure, whereas HMOX-1, an enzyme that eliminates oxidants [20], was induced at this timepoint; their expressions were all upregulated with repeated GA exposure. NQO1 is a flavoprotein that directly scavenges superoxide produced during oxidative stress as a superoxide reductase. GCS catalyzes the de novo synthesis of GSH, maintaining the intracellular GSH homeostasis. As an antioxidant protein, HMOX-1 has a more diverse panel of inducers than NQO1 and GCS. Based on the differential modulation of these proteins, we postulate that the immediate and more dominant effect of GA is to reduce the antioxidant levels by suppressing the expression or activity of the relevant enzymes; with time, the cellular system gradually adapts to the new oxidative environment and both antioxidative pathways function coordinately to establish a new homeostasis.

Mucociliary clearance (MCC) is one of the most important defense mechanisms of the human respiratory system. Disruptions in MCC is closely related with a multitude of chronic airway diseases [21]. Previous studies have reported that a single inhalation exposure of rats to 0.1 ppm GA damaged the cilia of airway bronchioles [22]. Our study consistently observed concentration-dependent decreases in CBF caused by single exposures to GA aerosols, accompanied by the downregulation of structural (i.e., acetylated α-tubulin) [23] and functional proteins (i.e., DNAI1) [24]. The direct interaction and crosslinking of proteins by GA may have accounted for the destructive effects of GA on the cellular apparatus that resides on the apical side. The histopathology data corroborated that the high GA concentration disrupted the apical structural and resulted in the loss of ciliated cells and the shrinkage of goblet cells. The induction of FoxJ1 expression, the master regulator of motile ciliogenesis [25], after a single exposure and the subsequent increase of CDC20B expression, a regulator of cilia production [26], at T5 suggest an attempt by the cells to promote ciliogenesis and ciliary differentiation. A previous study reported that a single 6-h inhalation exposure to 0.1 ppm GA reduced CCSP expression in the lung of rats [22]. The sustained impairment in the functional components of MCC observed in GA-exposed cultures may have a long-term impact on tissue function and respiratory disease development. Indeed, human case studies have consistently found that healthcare workers routinely working with GA are at a higher risk for obstructive respiratory diseases than the general population [27,28].

Occupational exposure to GA via inhalation has been implicated as the cause of respiratory sensitization in asthma and rhinitis [29,30,31]. The cytokine profile from cultured lymph nodes from mice treated topically with GA revealed its potential to elicit allergic sensitization of the respiratory tract [32]. In the current study, we found that repeated exposure of ALI cultures to GA aerosols induced dynamic changes in the secretion of select cytokines (IL-1RA, IL-6, IL-8, IL-9, FGF basic, G-CSF, GM-CSF, MCP-1, PDGF-BB, TNF-α, and VEGF). Consistent with our findings, previous studies have reported the increased production of IL-8 in bronchoalveolar lavage samples from patients with acute irritant-induced asthma [33]. Due to the limitation of this in vitro cell model, we were not able to show a T helper (TH2)-specific response, such as upregulated IgE levels and the induction of TH 2 type cytokines, as observed in GA-exposed mice [32,34]. As the ALI airway cultures lack the complexity of the in vivo immune system, most of the inflammatory responses observed in this study only reflect the immune regulation by epithelial cells; the impact of crosstalk between epithelial cells and other cell types, such as immune cells, fibroblasts, and endothelial cells, could not be assessed. 

Our histological observations on the degeneration of epithelial cells, as well as the significant loss of ciliated and goblet cells in the treated cultures, aligned with the respiratory irritant property of GA. Acute and subacute inhalation exposure to GA in rodents damaged ciliated cells and caused inflammation, degeneration, regeneration, goblet cell hyperplasia, necrosis, and squamous metaplasia of the respiratory tract [22,35,36]. GA is also an irritant to the skin, eyes, and respiratory system in humans, with effects similar to those found in animal studies [30,37]. 

Squamous metaplasia of the respiratory epithelium was observed in 13-week and 2-year inhalation studies of GA in mice [35,36]. In our in vitro study, we found that a 5-day repeated exposure of the ALI model to GA aerosols caused tissue damage with a prominent feature of squamous differentiation, as evidenced by both histological findings of typical columnar epithelial cells changing to atypical stratified squamous cells and the upregulation of involucrin expression, a marker for squamous metaplasia [38]. This morphological change was also corroborated by an increase in TEER values at both T5 and PT7, which implies that tissue integrity was enhanced, a feature consistent with the property of squamous epithelial cells. Compared to the pseudostratified respiratory epithelium, the squamous epithelium is more resistant to external damage. Thus, squamous differentiation can be seen as a tissue injury adaptation that may help preserve the damaged tissue, but at the cost of losing important tissue functions. Interestingly, IHC staining with involucrin suggests persistent, although to a lower degree, squamous differentiation at PT7; however, histopathology scoring of the H&E-stained tissue sections suggested otherwise. The discrepancy between these two methods could be due to the migration of involucrin from the cytosol to the inner surface of the membrane between T5 and PT7, leading to the formation of a complex between involucrin and the membrane proteins by transglutaminase [39].

Given that GA and OPA are both commonly used chemical disinfectants for heat-sensitive medical equipment, and OPA is considered a safer alternative to GA based on its lower bactericidal concentration, a comparison of respiratory toxicities between GA and OPA in the ALI airway system is, therefore, of interest. In our prior OPA study, repeated exposure to OPA aerosols (0.4, 0.6 and 1.0 µg/cm^2^) for 10 days caused a variety of functional and morphological changes [12] that were very similar to what were found in GA-exposed ALI cultures. Such concordance is consistent with that fact that both GA and OPA are reactive aldehydes with similar chemical properties, exerting similar biological effects, and having a common metabolic pathway [29,40]. It should be noted that the concentration range tested in the present study encompasses the concentrations used in the prior OPA study. Solely based on LDH release and CBF measurements, exposure to 1.0 µg/cm^2^ OPA for five days resulted in similar adverse responses (unpublished observations) when compared with exposure to 1.9 µg/cm^2^ of GA for the same exposure duration, suggesting that OPA may be a more potent respiratory tract irritant than GA in vitro. Consistent with our observations, the comparison of nasal and respiratory lesions in male rats exposed to GA or OPA by whole-body exposure for three months also revealed that 0.44 ppm OPA exposure resulted in a higher incidence of lesions with greater severity throughout the respiratory tract than did 0.50 ppm GA exposure [35,41], indicating that OPA may be a more potent respiratory toxicant than GA in vivo as well.

Considering the general concordance between our in vivo and in vitro findings, the human ALI airway model could be an important tool for evaluating and comparing the potential human risks posed by chemical exposures, such as GA and OPA. For the purposes of comparison, further studies with more concentration groups and employing the same exposure durations may better characterize their respiratory toxicity in a more quantitative way for the assessment of relative human risk.

Finally, findings from this and our prior case studies provide valuable scientific evidence supporting application of the human ALI airway model for toxicity testing of aerosolized chemicals as well as demonstrating that the ALI model may be useful in the CDRH Medical Device Development Tools (MDDT) program for use in the pre-market safety evaluations of future medical devices [42]. Once qualified for MDDT use, a major advantage of this in vitro non-clinical assessment model is the potential replacement for certain animal-based tests and a more streamlined pre-market review of relevant data. Stakeholders interested in modernizing the safety evaluation of medical devices with a potential for respiratory toxicity are hereby encouraged to help pool other ALI and/or complementary data to collaborate under the auspices of the MDDT program towards the qualification of this tool.

## 4. Materials and Methods

### 4.1. Glutaraldehyde (GA) Aerosol Deposition Quantification and Exposures in Air-Liquid-Interface (ALI) Cultures

GA (CAS No. 111-30-8) was obtained from Toronto Research Chemicals Inc. (North York, ON, Canada). Stock solutions were freshly prepared in diluted Dulbecco’s Phosphate-Buffered Saline (PBS; diluted at a ratio of 1:80 in molecular-grade water) to concentrations of 0.5, 0.75, and 1.0 mg/mL. The VITROCELL^®^ Cloud 12/12 Liquid Aerosol Exposure System (Waldkirch, Germany) was used for generating GA aerosols and exposing the ALI cultures at the air-liquid interface. Aerosol collection for analytical measurement and ALI cell exposures were carried out by following the procedures previously established in our laboratory [12]. Cultures were exposed to a bolus concentration of GA aerosols once a day for five consecutive days, followed by a 7-day recovery phase. 

To quantify the deposited aerosols in each insert, GA was derivatized with DNPH (Sigma-Aldrich, St. Louis, MO, USA) in 2 N HCl at room temperature for 30 min to form a stable derivative, GA-bis-DNPH. The derivative was extracted with dichloromethane and analyzed by a Waters ACQUITY UPLC system (Milford, MA, USA) coupled with a QDa mass detector. The analyte was eluted on an ACQUITY UPLC HSS T3 column (2.1 mm × 50 mm, 1.8 μm) at 40 °C using 10 mM ammonium acetate (Mobile phase A) and acetonitrile (Mobile phase B) both supplemented with 0.1% formic acid at a flow rate of 0.6 mL/min. The mobile phase was initially kept at 20% of B, followed by a 1.5-min linear gradient that finished at 95% of B. The composition of the mobile phase was restored to 20% of B in 0.1 min and balanced for 0.9 min. The eluate was detected by mass spectrometry equipped with an electrospray ion source operating in the negative ion mode (ESI^-^). A m/z of 459.2 was used to monitor GA-bis-DNPH. The amount of GA was quantified using a linear calibration curve that ranged from 0.31 μg/mL to 10 μg/mL and Waters Empower 3 software.

### 4.2. Human ALI Airway Tissue Model

Human ALI lower large airway tissue models were established using human primary tracheobronchial epithelial (NHBE) cells (MatTek, Ashland, MA, USA) and PneumaCult™ Culture Expansion and Differentiation Media (STEMCELL Technologies, Seattle, WA, USA) as described previously [9]. Cultures were differentiated by maintaining the air-liquid interface for at least five weeks (the day the cultures were air-lifted is day 0) before being used for experimentation. 

### 4.3. Lactate Dehydrogenase (LDH) Cytotoxicity Assay

The cytotoxicity of GA aerosols was assessed using a Lactate Dehydrogenase (LDH) Activity Assay Kit following the procedures recommended by the manufacturer (Roche, Indianapolis, IN, USA). The activity of released LDH was measured in both the apical wash (60 µL) and basolateral medium (100 µL) 24 h following one (T1) and five exposures (T5) as well as after a 7-day recovery (PT7). Apical washes were collected by washing the apical surface of the cultures with PBS (100 µL each wash for a total of two washes). The washes were combined and used for both LDH activity measurement and mucin secretion ELISA (described in the following section). Absorbance at 490 nm was measured using a Synergy H4 microplate reader (BioTek, Winooski, VT, USA).

### 4.4. Immunohistochemistry and Histology

Morphological changes caused by GA aerosols were evaluated at T5 and PT7. Sample preparation and histopathology staining were performed as described previously [12,13]. Hematoxylin and eosin (H&E) staining was conducted using a Leica Autostainer (Buffalo Grove, IL, USA); histopathologic changes were evaluated and scored by a board-certified pathologist. Digital images of the immunohistochemistry (IHC)-stained sections were obtained by scanning with the Aperio Scanscope System (Leica Biosystems, Vista, CA, USA). The percentages of cleaved caspase-3-positive apoptotic bodies, i.e., apoptotic index (AI), and periodic acid-Schiff (PAS)-stained goblet cells were calculated semi-automatically; the total numbers of nuclei were quantified automatically with the Nuclear Algorithm (Aperio Scanscope System), which counts the numbers of positive (brown) and negative (blue) nuclei in the section under examination, and the apoptotic bodies and PAS-stained goblet cells were counted manually. The percentage of Ki-67-positive nuclei, i.e., proliferative index (PI), was evaluated automatically with the Nuclear Algorithm.

### 4.5. Glutathione Homeostasis

Intracellular levels of reduced glutathione (GSH) and oxidized glutathione (GSSG) were quantified 20 min after the first exposure (20 min), and at T1, T5, and PT7 by using a liquid chromatography-mass spectrometry (LC-MS) method described previously [11]. 

### 4.6. Immunoblotting 

Expression of key biomarkers was assessed at T1, T5, and PT7 by immunoblotting. Cultures were lysed in a SIGMA*FAST*™-supplemented M-PER Mammalian Cell Lysis Buffer (Pierce, Waltham, MA, USA). Denatured proteins were separated on a 4–12% Nu-PAGE^®^ Novex^®^ Bis-Tris gradient gel (Life Technologies, Carlsbad, CA, USA) and transferred onto a nitrocellulose membrane (LI-COR, Lincoln, NE, USA) by following the manufacturer’s instructions. Primary antibodies used in this study included HMOX-1 (Cell Signaling Technology, Danvers, MA, USA), NQO1 (Santa Cruz Biotechnology, Dallas, TX, USA), AKR1B10 (Sigma-Aldrich), GCS (Santa Cruz Biotechnology), β-actin (Santa Cruz Biotechnology), acetylated α-tubulin (Sigma-Aldrich), DNAI1 (Sigma-Aldrich), CDC20B (Thermo Fisher Scientific, Waltham, MA, USA), Forkhead-box J1 (FoxJ1) (Santa Cruz Biotechnology), and involucrin (Neomarkers, Fremont, CA, USA). IRDye-conjugated secondary antibodies were obtained from LI-COR (Lincoln, NE, USA). The density of the protein bands was quantified using a LI-COR Odyssey imaging system and Image Studio Software (version 5.0). 

### 4.7. Bio-Plex Cytokine and Matrix Metalloproteinases (MMPs) Assays

The release of cytokines and MMPs into the basolateral medium was analyzed using a Bio-plex Pro Human cytokine 27-plex assay kit and a Human MMP 9-plex assay kit (Bio-Rad, Hercules, CA, USA), respectively, following the manufacturer’s instructions. Briefly, 50 µL basolateral media or analyte standards prepared in the PneumaCult™ Maintenance Medium were incubated with fluorescent magnetic beads for 1 h at 850 rpm ± 50 rpm in the dark. Unbound analytes were washed off by washing the plate three times with Wash Buffer. The analytes were then detected by incubating first with detection antibodies and then with Streptavidin-PE. The fluorescence of the beads was measured using a MAGPIX system (Luminex, Austin, TX, USA) and the intensity of the fluorescence was analyzed using the Bioplex Manager (Bio-Rad).

### 4.8. Cilia Beating Frequency (CBF)

CBF was measured at T1, T3, T5, and PT7 using the Sisson-Ammons Video Analysis System (SAVA System, Ammons Engineering, Clio, MI, USA). Briefly, cultures were equilibrated to 30 °C on a heated microscope stage before CBF measurement. The motility of the cilia was captured in four random fields devoid of mucus clumps using a high-speed camera (Ammons Engineering). Video data were analyzed automatically by the SAVA system to derive the CBF. 

### 4.9. Mucin ELISA

Secretion and expression of MUC5AC, MUC5B, and club-cell secretory protein (CCSP) were assessed in apical washes (50 µL) and cell lysates (5 µg), respectively, using a mucin ELISA method described previously [13]. 

### 4.10. Trans-Epithelial Electrical Resistance (TEER)

Tissue integrity was assessed by measuring TEER using an EVOM2 epithelial volt-ohmmeter and an STX2 chopstick electrode (World Precision Instruments, Sarasota, FL, USA) as described previously [12].

### 4.11. Statistical Analyses

Statistical analyses were performed using GraphPad Prism 7.01 (GraphPad Software, San Diego, CA, USA). Data from each timepoint were analyzed using one-way ANOVA, followed by Dunnett’s test for identifying treatment-related effects. *p* < 0.05 was considered statistically significant.

## Figures and Tables

**Figure 1 ijms-23-12118-f001:**
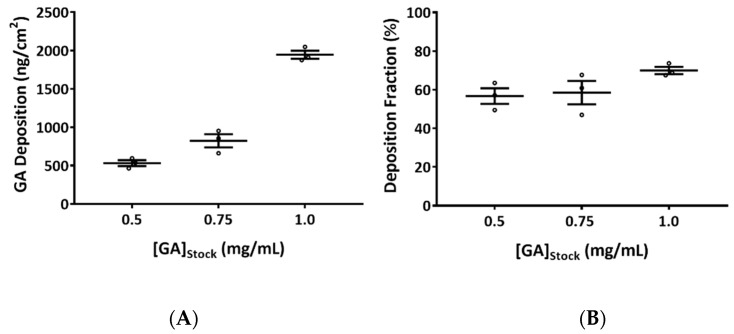
Quantification of glutaraldehyde (GA) aerosol deposition in the Cloud System. (**A**) Deposition of GA from 0.5, 0.75, and 1.0 mg/mL of the stock solutions was quantified using a UPLC method. Three independent nebulizations, each consisting of three representative positions, were conducted. Average deposition was calculated and presented. (**B**) Deposition fraction for each stock solution was graphed. Data (*n* = 3) are expressed as means ± standard error of the mean (SEM).

**Figure 2 ijms-23-12118-f002:**
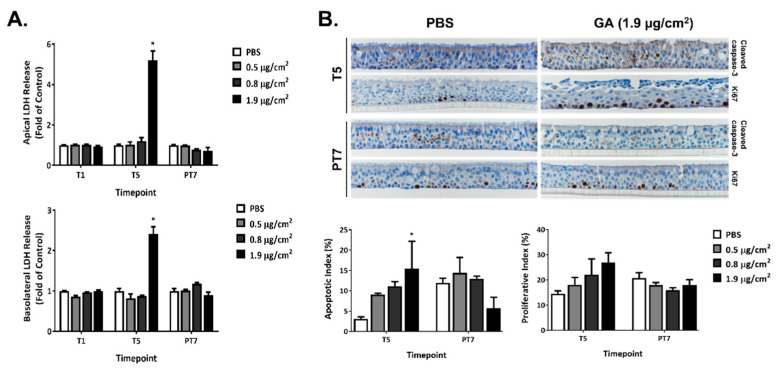
Cytotoxic effects of GA aerosols in the ALI cultures. (**A**) Cytotoxicity was assessed by measuring the LDH activity in both the apical washes and basolateral media at T1, T5, and PT7. (**B**) Apoptotic and proliferative effects of GA aerosols were evaluated by IHC with cleaved caspase-3 and Ki67 antibodies at T5 and PT7, respectively. Representative images (upper panel) were captured at 40× magnification. Apoptotic and proliferative indices were calculated and are presented in the bottom panel. Data (*n* = 3) are presented as means ± SEM. * *p* < 0.05 was considered statistically significant compared to the PBS-exposed vehicle controls.

**Figure 3 ijms-23-12118-f003:**
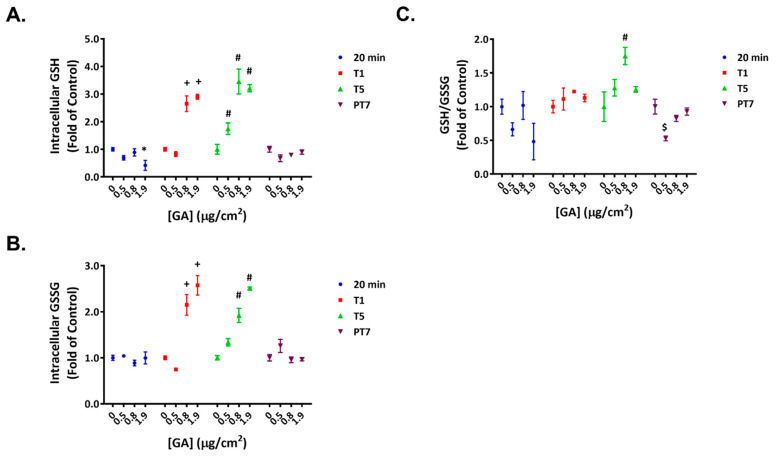
Temporary disruption of GSH/GSSG redox balance by GA aerosols. Intracellular levels of GSH (**A**) and GSSG (**B**) were quantified 20 min after the first exposure (20 min) and at T1, T5, and PT7. Normalized ratios of GSH/GSSG (**C**) were calculated for the respective timepoints. Data (*n* = 3) are presented as means ± SEM. *^,+,#,$^ *p* < 0.05 was considered statistically significant when compared to the respective vehicle controls.

**Figure 4 ijms-23-12118-f004:**
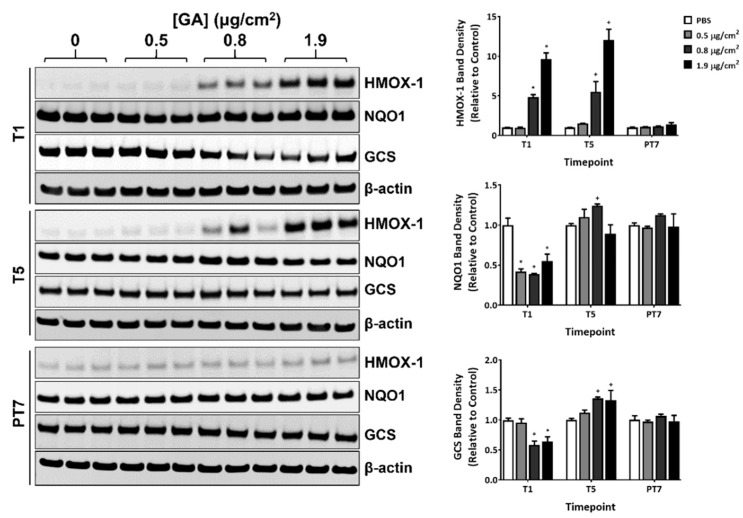
Modulation of the expression of oxidative stress markers by GA aerosols. Protein expression of HMOX-1, NQO1, and GCS was measured at T1, T5, and PT7 by immunoblotting. β-actin was used as the internal loading control for normalizing the expression of these proteins. Representative immunoblots are shown in the left panel. Quantification of the protein relative to the PBS-exposed vehicle controls is presented in the right panel. Data (*n* = 3) are expressed as means ± SEM. *^,+^ *p* <0.05 was considered statistically significant compared to the respective vehicle controls.

**Figure 5 ijms-23-12118-f005:**
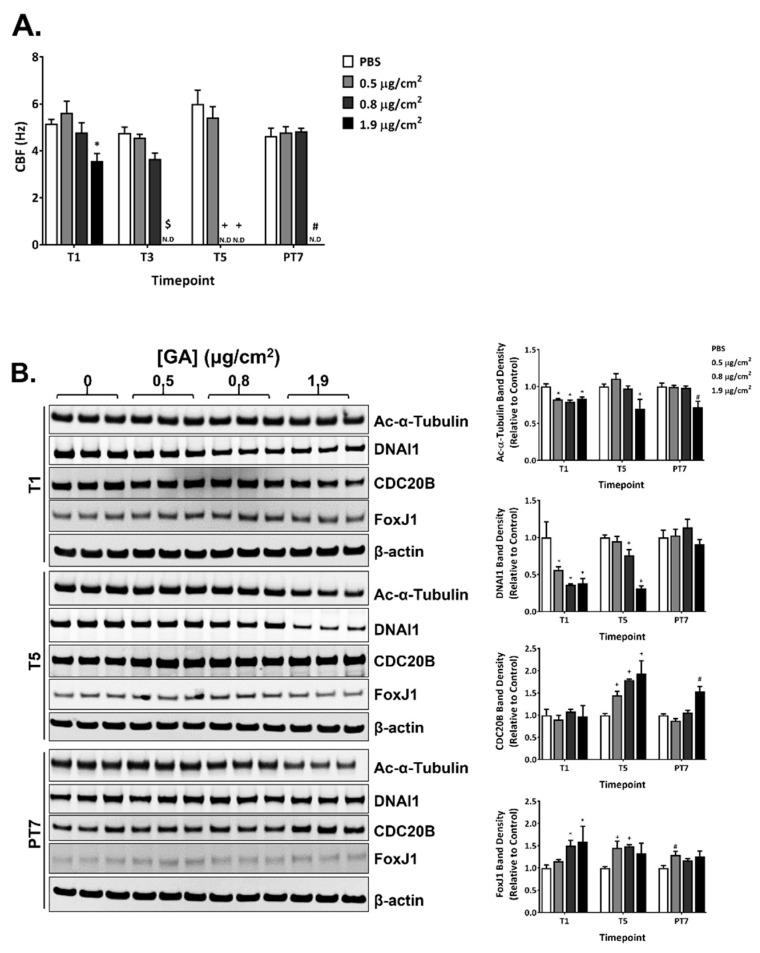
Inhibitory effects of GA aerosols on ciliary cells in the ALI cultures. CBF (**A**) was measured at T1, T3, T5, and PT7. (**B**) Expression of a panel of ciliary proteins was evaluated at T1, T5, and PT7 by immunoblotting. Representative images are shown in the left panel. The relative expression of these proteins is presented in the right panel. N.D: not detected. Data (*n* = 3) are expressed as means ± SEM. *^,+,$,#^ *p* < 0.05 was considered statistically significant compared to the respective vehicle controls.

**Figure 6 ijms-23-12118-f006:**
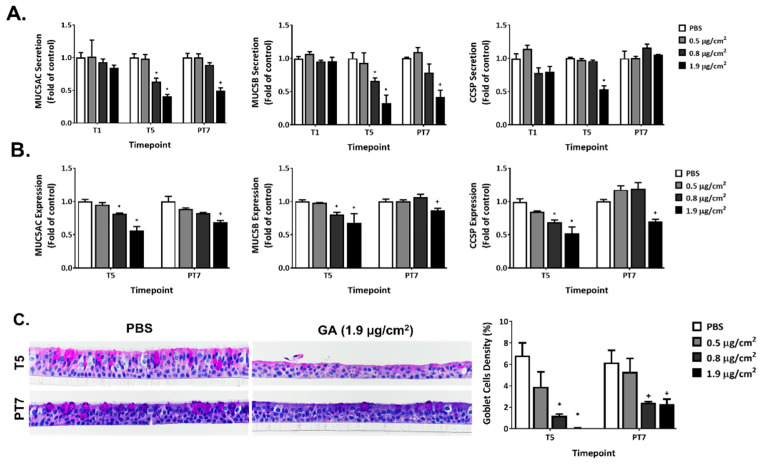
Modulation of GA aerosols on mucin homeostasis. Secretion (**A**) and expression (**B**) of MUC5AC, MUC5B, and CCSP were measured at T1 (only for mucin secretion), T5 and PT7. (**C**) Morphology and density of goblet cells were assessed at T5 and PT7 using periodic acid-Schiff (PAS) staining. Images were captured at 40× and presented in the middle panel. The percentage of goblet cells was calculated as described in the Materials and Methods section and is presented in the lower panel. Data (n = 3) are expressed as means ± SEM. *^,+^ *p* < 0.05 was considered statistically significant compared to the respective vehicle controls.

**Figure 7 ijms-23-12118-f007:**
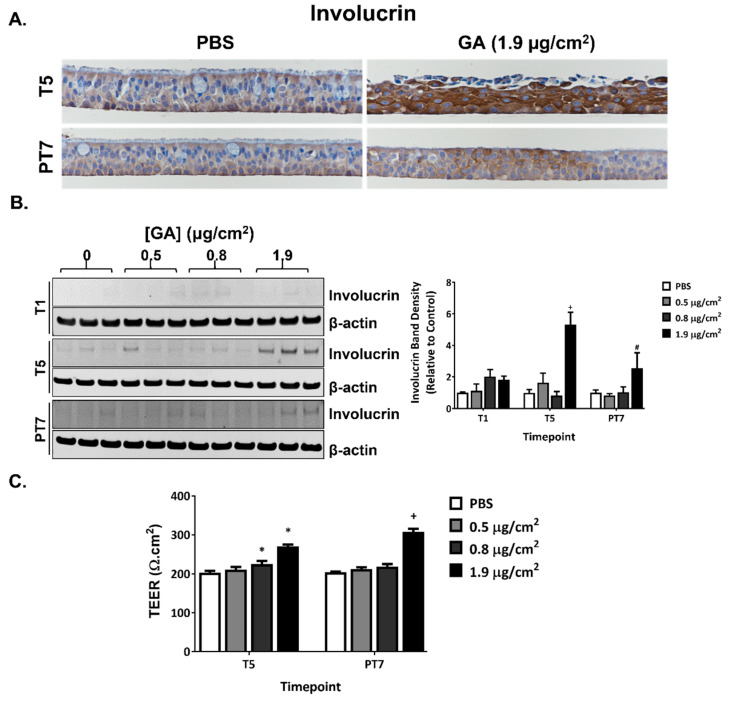
Squamous differentiation induced by GA aerosols in ALI cultures. (**A**) Induction of squamous differentiation was assessed by IHC staining with involucrin antibodies at T5 and PT7. Representative images were captured at a magnification of 40×. (**B**) The expression of involucrin was quantified at T5 and PT7 by immunoblotting. Representative blots were presented in the left panel. The quantification of the relative expression of involucrin is shown on the right panel. (**C**) TEER was measured at T5 and PT7. Data (*n* = 3) are presented as means ± SEM. *^,+,#^ *p* < 0.05 was considered statistically significant compared to the respective vehicle controls.

**Table 1 ijms-23-12118-t001:** Modulation of cytokine and MMP secretion by GA aerosols in ALI cultures.

Analytes (pg/mL)	T1				T5				PT7			
	PBS	0.5 µg/cm^2^	0.8 µg/cm^2^	1.9 µg/cm^2^	PBS	0.5 µg/cm^2^	0.8 µg/cm^2^	1.9 µg/cm^2^	PBS	0.5 µg/cm^2^	0.8 µg/cm^2^	1.9 µg/cm^2^
**IL-1RA**	49.2	48.5	44.2	47.4	151.8	170.5	165.4	194.5	195.6	206.0	299.1	345.0 *
	(8.0)	(5.1)	(8.1)	(5.2)	(17.9)	(13.9)	(6.0)	(24.3)	(19.8)	(22.0)	(62.60	(61.9)
**IL-6**	3.4	2.7	2.7	5.2 *	4.4	3.2	6.0	6.4	18.8	8.5	5.7	4.0 *
	(0.2)	(0.4)	(0.4)	(0.6)	(0.7)	(1.1)	(1.5)	(3.7)	(5.9)	(1.2)	(0.5)	(1.1)
**IL-8**	4911.2	3784.9	3734.8	8713.9 *	6822.0	5743.5	6669.1	11,908.0 *	14,452.8	11,649.5	8074.2 *	8825.2 *
	(772.8)	(442.6)	(178.7)	(1324.2)	(977.6)	(308.7)	(473.6)	(816.9)	(1856.3)	(591.1)	(2185.7)	(2354.5)
**IL-9**	34.6	30.3	31.9	44.0 *	38.3	35.9	38.6	50.4 *	56.8	51.5	42.9 *	43.8 *
	(2.1)	(2.8)	(3.0)	(1.9)	(2.2)	(2.2)	(0.9)	(0.9)	(2.1)	(2.1)	(6.7)	(3.7)
**FGF basic**	264.6	247.7	265.1	208.1	260.3	257.0	202.0	218.7	293.5	277.9	230.0	95.5 *
	(28.1)	(38.4)	(47.3)	(16.8)	(29.3)	(53.0)	(51.6)	(11.9)	(75.8)	(27.6)	(52.7)	(49.3)
**G-CSF**	67.0	52.8	47.0	71.4	76.2	61.0	92.3	83.2	268.7	172.8	114.6 *	142.3 *
	(3.7)	(18.1)	(17.4)	(31.4)	(24.3)	(15.2)	(12.7)	(38.7)	(54.4)	(52.0)	(25.3)	(35.2)
**GM-CSF**	6.0	5.1	4.3	4.3	8.3	6.3	6.7	8.9	17.1	12.1 *	8.6 *	6.2 *
	(0.9)	(1.3)	(0.2)	(0.8)	(1.9)	(0.7)	(1.2)	(0.8)	(1.4)	(1.4)	(0.5)	(0.8)
**MCP-1**	27.1	26.5	23.5	22.9	43.4	35.2	46.3	44.8	131.4	90.1	69.3 *	53.8 *
	(3.5)	(5.1)	(0.8)	(4.5)	(7.0)	(3.9)	(11.7)	(6.9)	(16.0)	(28.5)	(11.6)	(6.9)
**PDGF-BB**	986.6	1080.7	958.6	651.9 *	1473.7	1502.4	1457.9	1625.9	2425.6	2510.1	2339.1	2131.5 *
	(48.5)	(22.5)	(48.5)	(66.6)	(50.9)	(114.4)	(66.7)	(63.5)	(98.4)	(90.1)	(94.0)	(66.5)
**TNF-α**	46.3	47.2	39.6	43.5	73.8	68.6	63.9	79.8	180.0	146.8	102.2 *	84.7 *
	(1.6)	(4.3)	(5.7)	(1.6)	(9.0)	(3.4)	(7.7)	(5.8)	(5.9)	(7.3)	(3.8)	(7.9)
**VEGF**	628.0	584.8	578.1	706.1	637.8	632.6	571.7	489.6 *	1301.7	1266.8	1224.2	1126.9 *
	(35.6)	(32.6)	(26.8)	(48.0)	(32.4)	(47.6)	(21.5)	(72.2)	(35.6)	(92.7)	(29.7)	(66.3)
**MMP-1**	919.4	959.5	939.8	1000.1	1146.6	1059.1	1117.6	1242.0	1868.1	1559.8	1416.6 *	1295.8 *
	(35.4)	(70.2)	(0.0)	(0.0)	(33.3)	(48.1)	(54.8)	(234.1)	(233.0)	(91.5)	(83.0)	(168.7)
**MMP-7**	767.8	696.8	713.4	882.5	1493.0	1120.4	1696.3	1885.8	8043.5	4890.5 *	2744.9 *	2110.0 *
	(74.4)	(51.1)	(45.3)	(205.9)	(330.3)	(67.2)	(689.1)	(564.2)	(1977.4)	(731.8)	(465.6)	(618.8)
**MMP-12**	56.9	50.4	77.4 *	63.9	122.5	117.0	175.7 *	180.1 *	608.8	299.9 *	191.9 *	160.5 *
	(2.1)	(5.8)	(14.0)	(6.7)	(16.4)	(10.2)	(18.9)	(23.9)	(80.3)	(37.9)	(21.1)	(49.1)

* *p* < 0.05 compared to the respective PBS-exposed vehicle control; data are expressed as mean (SEM).

**Table 2 ijms-23-12118-t002:** Morphological changes in GA-exposed ALI cultures at T5 and PT7.

	T5				PT7			
	DPBS	0.5 µg/cm^2^	0.8 µg/cm^2^	1.9 µg/cm^2^	DPBS	0.5 µg/cm^2^	0.8 µg/cm^2^	1.9 µg/cm^2^
Atrophy	0/3	0/3	0/3	3/3	0/3	0/3	0/3	2/3
	(0.0)	(0.0)	(0.0)	(1.3)	(0.0)	(0.0)	(0.0)	(1.0)
	[0%]	[0%]	[0%]	[100%]	[0%]	[0%]	[0%]	[67%]
Ciliation, Decreased	0/3	0/3	3/3	3/3	0/3	0/3	1/3	3/3
	N/A	N/A	N/A	N/A	N/A	N/A	N/A	N/A
	[0%]	[0%]	[100%]	[100%]	[0%]	[0%]	[33%]	[100%]
Depletion, Goblet Cell	0/3	2/3	3/3	3/3	0/3	0/3	0/3	1/3
	(0.0)	(1.0)	(1.7)	(2.3)	(0.0)	(0.0)	(0.0)	(1.0)
	[0%]	[67%]	[100%]	[100%]	[0%]	[0%]	[0%]	[33%]
Differentiation, Squamous	0/3	0/3	0/3	2/3	N/D	N/D	N/D	N/D
	(0.0)	(0.0)	(0.0)	(1.5)				
	[0%]	[0%]	[0%]	[67%]				

Note: Data in the table are presented as follows: number of the ALI cultures affected/number of the ALI cultures examined; (average severity of affected ALI cultures rounded to the nearest tenth of a whole number); [percent incidence rounded to a percentage as a whole number]. N/A: not applicable; N/D: not detected.

## Data Availability

Not applicable.

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
