# Peer review of "Subacute Pulmonary Toxicity of Glutaraldehyde Aerosols in a Human In Vitro Airway Tissue Model"

_ijms, 2022, doi:10.3390/ijms232012118_

Round 1

Reviewer 1 Report

This is a well-planned and well-presented study assessing Glutaraldehyde (GA) toxicity.  A liquid aerosol exposure system was used to evaluate tissue responses in an ALI model to exposure. A panel of temporal changes in ALI tissue responses was documented after repeated exposure to GA aerosols and certain recovery periods. The molecular mechanisms underlying the toxicity were also explored. The results are a valuable addition to the documentation, as well as the understanding, of the respiratory toxicity of airborne chemicals. I recommend the publication of this manuscript after addressing some minor concerns, as listed below:

Lines 68-73: this is a long sentence and difficult to read. Please break it into several sentences.

Figure 1: please rearrange A, B in parallel instead of upper and down.

There are empty spaces at the bottom of pages 3 & 5. Need to rearrange the text and figures to remove/minimize these fly spaces.

Author Response

We would like to thank the reviewers for their review and constructive suggestions. All comments have been addressed point-by-point to the best of our knowledge; changes made in the revised manuscript are highlighted in yellow.

Reviewer 1

This is a well-planned and well-presented study assessing Glutaraldehyde (GA) toxicity.  A liquid aerosol exposure system was used to evaluate tissue responses in an ALI model to exposure. A panel of temporal changes in ALI tissue responses was documented after repeated exposure to GA aerosols and certain recovery periods. The molecular mechanisms underlying the toxicity were also explored. The results are a valuable addition to the documentation, as well as the understanding, of the respiratory toxicity of airborne chemicals. I recommend the publication of this manuscript after addressing some minor concerns, as listed below:

Comment 1: Lines 68-73: this is a long sentence and difficult to read. Please break it into several sentences.

Response: This sentence has been broken into 2 sentences. The changes are highlighted in yellow.

Comment 2: Figure 1: please rearrange A, B in parallel instead of upper and down.

Response: The format of Figure 1 has been re-arranged.

Comment 3: There are empty spaces at the bottom of pages 3 & 5. Need to rearrange the text and figures to remove/minimize these fly spaces.

Response: By re-arranging Figure 1, the empty space on page 3 is gone. We have moved the legend for Figure 3 next to Figure 3B.

Reviewer 2 Report

The paper by Wang et al. investigates the effects of aerosol exposure of various concentrations of GA in a in vitro human airway epithelial tissue model grown at the air interface, evaluating different parameters (oxidative stress, inhibition of ciliary beating frequency, aberrant mucin production, cytokine and matrix metalloproteinase secretion, and  morphological transformation) after 5 days of consecutive treatment and after a 7-day recovery phase.  The authors finally stated that the human air-liquid-interface (ALI) airway tissue  model, integrated with an in vitro exposure system simulating human inhalation exposure, could be used for hazard and risk characterization of aerosolized chemicals.

 In general, the research design is appropriate with adequate description of the methods and presentation of the results, anyway I have some minor suggestions for the authors :

 In my opinion in the discussion you should better detail the physiological role of NQO1, GCS and HMOX-1 to better clarify the different effect of GA treatment between the first two enzymes and the last one.

 Moreover, a possible explanation of TEER increase after GA treatment should be provided in the discussion, even considering the fact that TEER remains significantly higher at both T5 and PT7 with the maximum dose used.

 The size of the bottom panels of fig. 2B (apoptotic and proliferative index) and of the four right panels with histograms of fig. 5B should be slightly increased

 Figure 7A should be fixed since there is a floating title : (GA 1.9

Author Response

We would like to thank the reviewers for their review and constructive suggestions. All comments have been addressed point-by-point to the best of our knowledge; changes made in the revised manuscript are highlighted in yellow.

Review 2

The paper by Wang et al. investigates the effects of aerosol exposure of various concentrations of GA in a in vitro human airway epithelial tissue model grown at the air interface, evaluating different parameters (oxidative stress, inhibition of ciliary beating frequency, aberrant mucin production, cytokine and matrix metalloproteinase secretion, and  morphological transformation) after 5 days of consecutive treatment and after a 7-day recovery phase.  The authors finally stated that the human air-liquid-interface (ALI) airway tissue  model, integrated with an in vitro exposure system simulating human inhalation exposure, could be used for hazard and risk characterization of aerosolized chemicals.

 In general, the research design is appropriate with adequate description of the methods and presentation of the results, anyway I have some minor suggestions for the authors :

Comment 1: In my opinion in the discussion you should better detail the physiological role of NQO1, GCS and HMOX-1 to better clarify the different effect of GA treatment between the first two enzymes and the last one.

Response: The roles of NQO1, GCS and HMOX-1 have been added to the Discussion on pages 346 to 349 and is highlighted in yellow.

Comment 2: Moreover, a possible explanation of TEER increase after GA treatment should be provided in the discussion, even considering the fact that TEER remains significantly higher at both T5 and PT7 with the maximum dose used.

Response: This information was added to the Discussion on pages 402 to 404 and is highlighted in yellow.

Comment 3: The size of the bottom panels of fig. 2B (apoptotic and proliferative index) and of the four right panels with histograms of fig. 5B should be slightly increased

Response: The size of the quantification graph in Figure 2 has been enlarged. Histograms on the densitometry data in Figure 5 have been enlarged as well.

Comment 4: Figure 7A should be fixed since there is a floating title : (GA 1.9

Response: The figure with a floating title has been replaced with a complete figure in Figure 7.